# Prevalence and predictors of prediabetes/type 2 diabetes mellitus among adolescents in the United States: NHANES (2021–2023)

Eric Peprah Osei [1,2]*

1 College of Nursing, University of Illinois Chicago, Chicago, Illinois, United States of America, 2 Ghana Health Service, Dormaa West District Hospital, Bono Region, Ghana

* epepra2@uic.edu

## Abstract

Prediabetes and type 2 diabetes mellitus (T2DM) are emerging public health concerns among adolescents in the United States (U.S.), with early onset increasing the risk of lifelong complications. This study analyzed the prevalence and factors associated with prediabetes/T2DM among 1,998 adolescents (10–19 years) in the U.S. using data from the National Health and Nutrition Examination Surveys (2021–2023). Prediabetes/T2DM were defined as hemoglobin A1c (HbA1c) ≥ 5.7% or fasting plasma glucose (FPG) ≥ 100 mg/dL. Unweighted univariate and multiple logistic regression models were used to identify predictors of prediabetes/T2DM among adolescents. Overall, 30.8%—nearly 1 in 3 American adolescents—had prediabetes or T2DM. In univariate analysis, older age (OR = 0.93, p = 0.045), female gender (OR = 0.50, p = 0.001), overweight/obesity (OR = 1.57, p = 0.012), elevated waist-to-height ratio (OR = 24.04, p = 0.002), total daily sugar intake (OR = 1.003; p = 0.042), low HDL cholesterol (≤45 mg/dL) (OR = 1.41; p = 0.032), higher systolic blood pressure (OR = 1.02, p = 0.002) and higher diastolic blood pressure (OR = 1.02, p = 0.037) were significantly associated with the odds of having prediabetes/T2DM. However, in multiple logistic regression analysis, significant predictors included older age (AOR = 0.91; p = 0.025), female gender (AOR = 0.52; p = 0.002), and elevated waist-to-height ratio (AOR = 146.19; p = 0004). Although male gender and younger age showed increased risk, central adiposity—specifically measured by waist-to-height ratio—emerged as the strongest independent predictor of prediabetes/T2DM compared to general overweight/obesity (BMI). These findings underscore the need for early screening and targeted prevention strategies focusing on central adiposity and demographic risk factors.

**Data availability statement:** The data used in this study are publicly available from the National Health and Nutrition Examination Survey (NHANES) website at https://www.cdc.gov/nchs/nhanes/index.html. Researchers can access the 2021–2023 datasets used in this analysis following NHANES data use policies.

**Funding:** The author(s) received no specific funding for this work.

**Competing interests:** The authors have declared that no competing interests exist.

## Introduction

Over the past two decades, there has been a dramatic rise in the prevalence of prediabetes and type 2 diabetes mellitus (T2DM) among adolescents in the United States (U.S.) [1]. This trend is mainly driven by rising rates of obesity, sedentary lifestyle, and unhealthy dietary patterns [2,3]. The National Institute of Diabetes and Digestive and Kidney Diseases report in 2024 indicated that, the alarmingly high rate of T2DM disproportionately affects children and young people from racial and ethnic minority groups [4] and is projected to quadruple in the next four decades [5]. This evidence is particularly concerning given that previous studies have highlighted the more aggressive nature of youth-onset T2DM [6]. The associated complications such as diabetic kidney disease, cardiovascular disease, neuropathy and retinopathy [6,7] manifest earlier among adolescents due to rapid destruction of the pancreatic beta cells [8]. This negatively impacts their quality of life and subsequently leads to unfavorable long-term outcomes [8]. Concurrently, prediabetes, which is a key predictor of T2DM, is increasing rapidly among this vulnerable population [9]. This arguably portends an impending public health disaster and highlights a critical need for early identification and management.

Despite the increases in the burden of prediabetes and T2DM in the adolescent population, significant gaps persist in understanding the prevalence and risk factors for these conditions. Most epidemiological studies pertaining to T2DM have predominantly focused on older adults, leaving adolescents among the most understudied age brackets. A study conducted by Ouyang et. al. (2024) that analyzed National Health and Nutrition Examination Survey (NHANES) data (1999–2020) among adolescents only focused on obesity as a risk factor for diabetes and prediabetes [10]. However, the study did not thoroughly evaluate other predictors such as socioeconomic status, dietary patterns, waist-to-height circumference, physical activity and sedentary time. Furthermore, NHANES offers a nationally representative dataset to evaluate trends in prediabetes and T2DM across various age groups. Nevertheless, few studies have utilized the most up-to-date NHANES data from 2021 through 2023 in examining T2DM risks among adolescents, especially in the COVID-19 pandemic era, which further deteriorated metabolic health.

To address these gaps, this current study aimed to analyze the current NHANES (2021–2023) data to determine the prevalence of prediabetes/T2DM among adolescents (10–19 years) in the U.S. Additionally, the study sought to identify predictors of prediabetes/T2DM among adolescents in the U.S. Findings from the study are critical to inform public health policies and development of targeted interventions for this vulnerable population. The rising prevalence of early-onset T2DM in youth may cause a spike in diabetes-related complications if prompt interventions are not developed.

## Methods

### Ethics statement

This study used publicly available, de-identified data from NHANES (2021–2023). NHANES data collection was approved by the National Center for Health Statistics Research Ethics Review Board, and all participants provided informed consent.

Because the data are publicly available and de-identified, additional institutional review board approval was not required for this secondary analysis.

## Data source

This study conducted a secondary analysis of data from the 2021–2023 cycle of NHANES, a program conducted by the Centers for Disease Control and Prevention (CDC). NHANES uses a complex, multistage probability sampling method designed to produce a nationally representative sample of the U.S. civilian, non-institutionalized population. The survey collects detailed health and nutrition information through standardized in-home interviews, physical examinations, and laboratory assessments conducted at mobile examination centers.

## Study population

The study focused on adolescents aged 10–19 years. Participants with complete data on anthropometrics, fasting plasma glucose, and HbA1c were included. The final analytic sample consisted of 1,998 adolescents (1,001 males and 997 females). The dataset is de-identified and publicly available, negating the need for further ethical approval.

## Measurements and glucose status definitions

### Dependent variables

Although HbA1c and fasting plasma glucose (FPG) were originally reported as continuous variables, they were dichotomized to classify participants as having prediabetes/T2DM (1 = yes) and normal glucose tolerance (0 = no). Prediabetes/T2DM were defined as having HbA1c ≥ 5.7% or FPG ≥ 100 mg/dL whereas normal glucose levels were HbA1c < 5.7% or FPG < 100 mg/dL [11]. In the current study sample, 69.2% of participants had normal glucose, 30.5% had prediabetes, and 0.25% had T2DM. Due to the very low prevalence of T2DM (5 out of 1,998), separate modeling of prediabetes and T2DM would have resulted in unstable estimates and limited statistical power. Categorizing glucose status into "normal" and "prediabetes/T2DM "facilitates a more thorough and efficient analysis of the metabolic health of adolescents based on the risk factors and early intervention windows that are common to both disorders. This approach also maximizes statistical power and allows for data modeling.

### Independent variables

**Demographic variables.** Information about age, sex, race, and poverty income ratio (PIR) was obtained from demographic data. Age, a continuous variable, was dichotomized as early adolescents (10–14 years) and late adolescents (15–19 years). Sex was reported as a binary variable, representing male or female. There were six racial groups: Mexican American, other Hispanic, Non-Hispanic White, Non-Hispanic Black, Non-Hispanic Asian and other race. Ratio of family income to poverty level, a continuous variable, was defined as low income (PIR < 1.3), middle income (PIR ≥ 1.3 & < 3.5), and high income (PIR ≥ 3.5) [12]. Health insurance status was derived from the survey question, "Are you covered by health insurance or some other kind of health care plan?", with either 'Yes' or 'No' response.

**Anthropometric factors.** The body mass index (BMI) originally reported with four levels was further categorized as underweight/normal weight or overweight/obesity. Waist-to-height ratio, a measure of central obesity, was calculated by waist circumference (cm)/ height (cm) which was further dichotomized as healthy <0.5 or abdominal obesity ≥ 0.5 [13].

**Lifestyle factors.** Sedentary behavior was measured by hours per day spent watching TV or videos. This variable was further categorized into two groups of activities: < 2 hours/day and ≥2 hours/day [14]. Physical activity, measured as days physically active at least 60 min, was dichotomized as those who met physical activity guidelines (≥60 minutes/day on all 7 days) and those that did not meet guidelines (<60 minutes on any day) [15].

**Dietary patterns.** Dietary consumption patterns were recorded through 24-hour dietary recall interviews. Total sugar intake and total carbohydrate intake, measured in grams, were treated as continuous variables. Energy intake (kcal) was

categorized based on the Dietary Guidelines for Americans, 2020–2025, which recommend 1,400–2,200 kcal/day for females aged 9–13 years, 1,800–2,400 kcal/day for females aged 14–18 years, 1,600–2,600 kcal/day for males aged 9–13 years, and 2,000–3,200 kcal/day for males aged 14–18 years [16]. Participants were classified as consuming (1) below, (2) within, or (3) above the recommended intake for their age and sex.

**Clinical and biochemical factors.** Total cholesterol, high-density lipoprotein cholesterol (HDL-C) and C-reactive protein values were obtained from the laboratory data. Healthy children, 19 years old and younger, should have: total cholesterol below 170 mg/dL and high-density lipoprotein (HDL) above 45 mg/dL [17]. Likewise, C-reactive protein (hs-CRP) value of >3.0 mg/L was considered a marker for inflammation and elevated risk for CVD [18,19]. For adolescents aged 10–12 years: Normal BP: <90th percentile; Elevated BP: ≥90th to <95th percentile or 120/80 mmHg to <95th percentile (whichever is lower); Hypertension: ≥95th percentile to <95th percentile + 12 mmHg or ≥130/80 mmHg (whichever is lower). For adolescents aged ≥13 years: Normal BP: <120/<80 mmHg; Elevated BP: 120/<80–129/<80 mmHg; Hypertension: ≥130/80 mmHg. [12].

### Data analysis

Data analysis was performed using STATA software, version 18. Descriptive statistics comprised means, standard deviations, frequencies, and percentages that determined not only the prevalence of prediabetes/T2DM but also characterized the study population (adolescents). To assess the robustness of the findings, sensitivity analyses were conducted using the fasting subsample (n = 571) with appropriate NHANES fasting subsample weights (wtsaf2yr), accounting for strata and primary sampling units (sdmvstra, sdmvpsu). For the second aim, multiple logistic regression models were employed to examine potential predictors of prediabetes/T2DM among adolescents. Furthermore, sequential logistic regression models were fitted to examine the association between overweight/obesity and abnormal glucose status (S2 Table). Missing data were handled by multivariate multiple imputation using chained equations to minimize bias and ensure the robustness of the findings.

### Results

This study analyzed data from 1,998 U.S. adolescents (10–19 years) from the National Health and Nutrition Examination Surveys (NHANES, 2021–2023) to determine the prevalence and predictors of prediabetes/type 2 diabetes mellitus. The analysis included 1,998 U.S. adolescents (mean age 14.2 ± 2.8 years; 50.1% male). Overall, 41.2% were overweight/obese and 40.8% had abdominal obesity (waist-to-height ratio ≥0.5). Sedentary behavior was highly prevalent (88.5% ≥ 2 h/day), while only 20.7% met physical activity guidelines. The mean fasting plasma glucose (FPG) and HbA1c were 95.2 mg/dL and 5.25%, respectively, with males exhibiting higher mean FPG (96.80 vs. 93.68 mg/dL). Systolic blood pressure and HDL cholesterol levels differed by sex, with males having higher mean systolic BP (109.5 vs. 103.7 mmHg) and lower mean HDL (50.5 vs. 54.4 mg/dL). Total cholesterol and C-reactive protein levels were similar between both sexes (Table 1).

Table 2 presents the unweighted prevalence of prediabetes/T2DM among adolescents aged 10–19 years, stratified by key demographic and health variables. The overall weighted prevalence of prediabetes/T2DM was a concerning 30.8%, meaning nearly 1-in-3 American adolescents has the condition (Fig 1). The prevalence of prediabetes/ T2DM was significantly higher in males (62.0%) compared to females (38.0%). Non-Hispanic White adolescents (37.2%) had the highest rates across racial/ethnic groups. Adolescents with overweight/obesity (48.8%) and those with abdominal obesity (waist-to-height ratio ≥ 0.5) (48.7%) had prevalence rates similar to their healthier-weight counterparts.

### Logistic regression model assumptions

Prior to model fitting, assumptions underlying logistic regression were evaluated. To assess multicollinearity among the independent variables, variance inflation factors (VIF) were examined. Total carbohydrate intake (tcarb) showed a VIF of 10.25, indicating high multicollinearity, and was therefore removed from the model. All remaining predictors had VIF

**Table 1. Demographics and key characteristics of adolescents (10-19 years, n=1998).**

| Variable | Total Sample n=1998 | Males n=1001 | Females n=997 |
|---|---|---|---|
| **Age (years), n (%)** | | | |
| Mean | 14.2±2.80 | 14.2±2.78 | 14.2±2.83 |
| Early adolescents (10–14) | 1088 (54.5) | 542 (54.1) | 546 (54.8) |
| Late adolescents (15–19) | 910 (45.5) | 459 (45.9) | 451 (45.2) |
| **Race/Ethnicity, n (%)** | | | |
| Mexican American | 305 (15.3) | 141 (14.1) | 164 (16.4) |
| Other Hispanic | 282 (14.1) | 147 (14.7) | 135 (13.6) |
| Non-Hispanic White | 785 (39.3) | 380 (37.9) | 405 (40.6) |
| Non-Hispanic Black | 302 (15.1) | 151 (15.1) | 151 (15.2) |
| Non-Hispanic Asian | 125 (6.3) | 68 (6.8) | 57 (5.7) |
| Other Race | 199 (9.9) | 114 (11.4) | 85 (8.5) |
| **Health Insurance Status (Yes), n (%)** | 1876 (93.9) | 930 (92.9) | 946 (94.8) |
| **Poverty income ratio (PIR), n (%)** | | | |
| Low income (PIR<1.3) | 687 (34.4) | 333 (33.3) | 354 (35.5) |
| Middle income (PIR≥1.3 & <3.5) | 804 (40.3) | 407 (40.6) | 398 (39.9) |
| High income (PIR≥3.5) | 507 (25.3) | 261 (26.1) | 246 (24.6) |
| **BMI Categories, n (%)** | | | |
| Mean | 23.76±7.35 | 23.62±.23 | 23.91±.24 |
| Underweight/Normal Weight | 1174 (58.8) | 587 (58.7) | 587 (58.9) |
| Overweight/Obesity | 824 (41.2) | 414 (41.3) | 410 (41.1) |
| **Waist-to-Height Ratio, n (%)** | | | |
| Healthy <0.5 | 1183 (59.2) | 614 (61.3) | 569 (57.1) |
| Abdominal obesity ≥0.5 | 815 (40.8) | 387 (38.7) | 428 (42.9) |
| **Total sedentary time, n (%)** | | | |
| <2hours/day | 231 (11.5) | 123 (12.3) | 107 (10.8) |
| ≥2hours/day | 1767 (88.5) | 878 (87.7) | 890 (89.2) |
| **Days Physically Active, n (%)** | | | |
| Did not meet guidelines | 1584 (79.3) | 755 (75.4) | 829 (83.1) |
| Met guidelines (≥60 Minutes * 7days) | 414 (20.7) | 246 (24.6) | 168(16.9) |
| HbA1c (%), mean±SD | 5.25±.41 | 5.25±.48 | 5.24±.40 |
| FPG (mg/dl), mean±SD | 95.24±15.71 | 96.80±20.62 | 93.68±13.49 |
| Systolic BP, mean±SD | 106.6±10.34 | 109.48±10.46 | 103.71±9.66 |
| Diastolic BP, mean±SD | 63.77±8.50 | 64.0±9.20 | 63.54±8.20 |
| Total cholesterol, mean±SD | 155.36±38.38 | 154.30±37.81 | 156.43±39.94 |
| HDL cholesterol, mean±SD | 52.42±16.31 | 50.45±12.51 | 54.40±15.93 |
| C-reactive protein, mean±SD | 1.99±5.47 | 1.98±8.09 | 1.99±8.09 |

n (%) – Frequency (percentage); SD – Standard deviation; FPG – Fasting plasma glucose; HbA1c – Glycated hemoglobin; BMI – Body mass index; HDL – High-density lipoprotein; BP – Blood pressure.

values below 10, suggesting no significant collinearity issues. The assumption of linearity in the logit was tested for continuous variables. Several variables including poverty-income ratio (indfmpir), total cholesterol (lbxtc), HDL cholesterol (lbdhdd), BMI (bmxbmi), sedentary activity (paq706), physical activity (paq711), and total energy intake (tkcal), violated this assumption. Consequently, these variables were categorized based on established criteria in literature.

**Table 2. Unweighted prevalence of prediabetes/T2DM among adolescents (10-19 years, n = 1,998), Stratified by Key Variables.**

| Variable | Normal glucose Unweighted % (95% CI) | Prediabetes/ T2DM Unweighted % (95% CI) | p-value |
|---|---|---|---|
| Overall Prevalence | 69.2 [65.5 - 73.0] | 30.8 [27.0 - 34.5] | – |
| Gender | | | 0.000[a] |
| Male | 44.8 [41.5-48.1] | 62.0 [56.4-67.6] | |
| Female | 55.2 [51.9-58.5] | 38.0 [32.4-43.6] | |
| Race/Ethnicity | | | 0.500[b] |
| Mexican American | 14.6 [12.3-17.0] | 16.7 [12.2-21.1] | |
| Other Hispanic | 13.6 [11.5-15.7] | 15.3 [11.5-19.0] | |
| Non-Hispanic White | 40.2 [37.1-43.2] | 37.2 [31.9-42.6] | |
| Non-Hispanic Black | 15.4 [13.2-17.7] | 14.4 [10.6-18.2] | |
| Non-Hispanic Asian | 6.5 [5.0-8.0] | 5.7 [3.2-8.3] | |
| Other Race | 9.7 [7.7-11.7] | 10.7 [6.9-14.5] | |
| BMI Categories | | | 0.000[a] |
| Underweight/ healthy weight | 62.2 [58.8-65.5] | 51.2 [44.3-58.1] | |
| Overweight/ obesity | 37.8 [34.5-41.2] | 48.8 [41.9-55.7] | |
| Waist-to-Height Ratio | | | 0.000[a] |
| Healthy (<0.5) | 62.7 [59.5 – 66.0] | 51.3 [44.5 – 58.2] | |
| Abdominal obesity (≥ 0.5) | 37.3 [34.0 – 40.5] | 48.7 [41.8 – 55.5] | |
| Poverty income ratio (PIR) | | | 0.722[b] |
| Low income (PIR < 1.3) | 33.8 [30.9-36.8] | 35.5 [30.1-41.0] | |
| Middle income (PIR ≥ 1.3 & < 3.5) | 40.8 [37.5-44.1] | 39.0 [33.7-44.3] | |
| High income (PIR ≥ 3.5) | 25.3 [22.5-28.2] | 25.5 [20.5-30.4] | |
| Health Insurance Status | | | 0.920[a] |
| Yes | 93.9 [92.4 – 95.3] | 93.9 [91.4 – 96.4] | |
| No | 6.1 [4.7 – 7.6] | 6.1 [3.6 – 8.6] | |

[a]Fisher's exact test; [b] Pearson chi-square; BMI -Body mass index. Although NHANES survey weights were available, unweighted multiple-imputation estimates are reported for all participants (n = 1,998) to maximize statistical power. Weighted analyses restricted to the fasting subsample (n = 571) that accounted for survey weights, strata, and primary sampling units produced comparable findings (S1 Table).

## Logistic regression analysis of factors associated with prediabetes/T2DM

Table 3 presents the univariate and multivariate logistic regression models exploring associations between demographic, anthropometric, lifestyle, dietary, and clinical factors with prediabetes/T2DM among adolescents aged 10–19 years. Univariate analysis identified several significant predictors of prediabetes/T2DM. Lower odds were observed for older age (OR = 0.93, 95% CI: 0.87–0.99) and female gender (OR = 0.50, 95% CI: 0.36–0.68). Conversely, higher odds were associated with overweight/obesity (OR = 1.57, 95% CI: 1.11–2.21), elevated waist-to-height ratio (OR = 24.04, 95% CI: 3.62–159), higher daily sugar intake (OR = 1.003, 95% CI: 1.000–1.005), low HDL cholesterol (≤45 mg/dL) (OR = 1.41, 95% CI: 0.79–1.65), and higher systolic (OR = 1.02, 95% CI: 1.01–1.03) and diastolic blood pressure (OR = 1.02, 95% CI: 1.00–1.04).

Multivariate analysis (after adjusting for confounders) identified three independent, significant predictors. Elevated waist-to-height ratio (central adiposity) emerged as the strongest independent predictor, with adolescents having over 146 times higher odds (AOR = 146.19, 95% CI: 5.39, 3976). BMI status, in contrast, lost significance in the multivariate model.

 

PLOS Global Public
Health

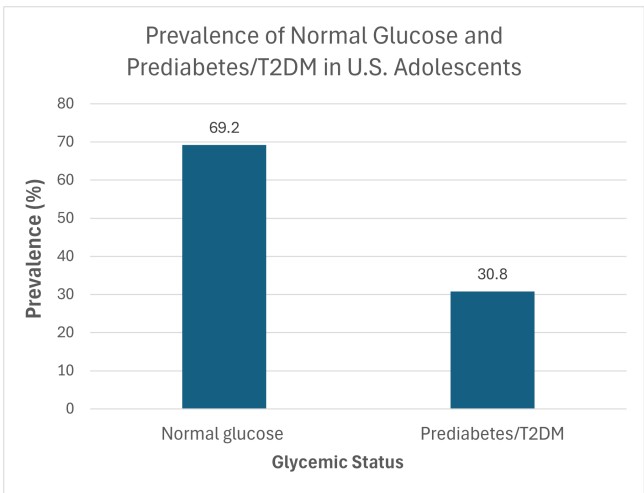

**Fig 1.** **Prevalence of normal glucose vs prediabetes/T2DM in U.S. adolescents (*Data source: NHANES 2021–2023*).**

Female gender was associated with lower odds (AOR = 0.52, 95% CI: 0.36, 0.78) compared to males. Older age was associated with lower odds (AOR = 0.91, 95% CI: 0.83, 0.99) of prediabetes/T2DM in adolescents.

## Sensitivity analysis

Sensitivity analyses were conducted by sequentially adding covariates to examine the robustness of the association between overweight/obesity and abnormal glucose status. In the unadjusted model (Model 1), overweight/obesity was significantly associated with higher odds of abnormal glucose status (OR = 1.45, 95% CI: 1.10–1.79). After adjusting for socio-demographics and waist-to-height ratio (Model 2), the association was attenuated and became non-significant (OR = 0.76, 95% CI: 0.44–1.32). Further adjustment for physical activity and sedentary behavior (Model 3) yielded a similar non-significant result (OR = 0.77, 95% CI: 0.45–1.33). Finally, including dietary intake and metabolic markers (Model 4) did not change the pattern (OR = 0.74, 95% CI: 0.43–1.29), while age (OR = 0.91, 95% CI: 0.83–0.99), female sex (OR = 0.53, 95% CI: 0.36–0.78), and waist-to-height ratio (OR = 146.42, 95% CI: 5.39–3976.30) remained significant predictors. These results suggest that the observed relationship is influenced by demographic and metabolic factors, supporting the robustness of the study findings (S2 Table).

## Discussion

This analysis of nationally representative NHANES data demonstrates a concerning prevalence of prediabetes/T2DM (30.8%) among U.S. adolescents, supporting previous evidence that abnormal glucose regulation is a major emerging public health issue in youth [1,20]. A critical observation is the disproportionate burden among males, who accounted for 62% of cases, compared to 38% among females. This sex disparity persisted even after multivariate adjustment, suggesting potential biological underpinnings such as greater visceral fat accumulation [21] and androgen-mediated insulin resistance [22]. This pattern aligns with U.S. and international studies [1,23,24] and a meta-analysis [25] reporting higher prediabetes/ T2DM prevalence in adolescent males, highlighting the potential importance of sex-specific preventive strategies.

One of the most striking findings is the strong predictive power of waist-to-height ratio for prediabetes/T2DM, consistent with global evidence showing high rates of overweight and obesity among youth with type 1 diabetes mellitus [26]. While overweight/obesity by BMI was associated with prediabetes/T2DM in univariate analysis, the association was attenuated

**Table 3. Logistic regression of factors associated with prediabetes/T2DM in adolescents.**

| Variable | Univariate | | Multivariate | |
|---|---|---|---|---|
| | Unadjusted OR (95% CI) | p-value | Adjusted OR (95% CI) | P value |
| *Demographic variables* | | | | |
| Age (years) | 0.93 [0.87, 0.99] | 0.045 | 0.91 [0.83, 0.99] | 0.025 |
| Gender (Female) | 0.50 [0.36, 0.68] | 0.000 | 0.53 [0.36, 0.78] | 0.002 |
| Race (Ref: Other Race) | | | | |
| Mexican American | 1.03 [0.53, 2.02] | 0.925 | 1.16 [0.55, 2.43] | 0.686 |
| Other Hispanic | 1.02 [0.52, 2.01] | 0.957 | 1.13 [0.54, 2.35] | 0.738 |
| Non-Hispanic White | 0.84 [0.47, 1.50] | 0.549 | 0.95 [.051, 1.77] | 0.865 |
| Non-Hispanic Black | 0.84 [0.46, 1.55] | 0.579 | 0.95 [0.49, 1.83] | 0.867 |
| Non-Hispanic Asian | 0.79 [0.38, 1.68] | 0.548 | 0.94 [0.43, 2.07] | 0.884 |
| Poverty-income-ratio (Ref: Low income) | | | | |
| Middle income | 0.91 [0.65, 1.28] | 0.582 | 0.93 [0.64, 1.36] | 0.705 |
| High income | 0.95 [0.65, 1.40] | 0.816 | 1.09 [0.68, 1.74] | 0.717 |
| Health Insurance Status (Yes) | 1.01 [0.57, 1.80] | 0.964 | 1.04 [0.56, 1.95] | 0.891 |
| *Anthropometric factors* | | | | |
| BMI Status (Ref: Normal weight) | | | | |
| Overweight/Obesity | 1.57 [1.11, 2.21] | 0.012 | 0.74 [0.43, 1.29] | 0.285 |
| Waist-to-height ratio | 24.04 [3.62, 159] | 0.002 | 146.42 [5.39, 3976] | 0.004 |
| *Lifestyle factors* | | | | |
| Total sedentary time (Ref:<2hours/day) | | | | |
| ≥2hours/day | 0.77 [0.42, 1.42] | 0.395 | 0.80 [0.42, 1.52] | 0.493 |
| Physical Activity (Ref: Below Guidelines) | | | | |
| Met guidelines (≥60 Min * 7 days) | 1.17 [0.77, 1.79] | 0.453 | 1.07 [0.66, 1.71] | 0.784 |
| *Dietary factors* | | | | |
| Energy intake (k/Cal) (Ref: Below recommended) | | | | |
| Within recommended | 1.25 [0.89, 1.75] | 0.432 | 1.03 [0.68, 1.57] | 0.881 |
| Above recommended | 1.13 [0.69,1.87] | 0.161 | 0.93 [0.48, 1.81] | 0.833 |
| Total sugar intake (gm) | 1.003 [1.00, 1.005] | 0.042 | 1.00 [1.00, 1.01] | 0.228 |
| *Clinical and biochemical factors* | | | | |
| Total cholesterol (Ref: Normal) | | | | |
| Borderline/high (>=170mg/dL) | 1.14 [0.79, 1.65] | 0.463 | 1.07 [0.72, 1.59] | 0.738 |
| HDL cholesterol (Ref: Normal) | | | | |
| Low (<=45mg/dL) | 1.41 [1.03, 1.92] | 0.032 | 1.05 [0.75, 1.48] | 0.757 |
| C-reactive protein (3mg/L) | 1.00 [0.97, 1.03] | 0.997 | 0.99 [0.95, 1.02] | 0.439 |
| Systolic BP (mmHg) | 1.02 [1.01, 1.03] | 0.002 | 1.02 [0.98, 1.03] | 0.109 |
| Diastolic BP (mmHg) | 1.02 [1.00, 1.04] | 0.037 | 1.01 [0.98, 1.03] | 0.647 |

WHR: Waist-to-height ratio; OR: Odds ratio; CI: confidence interval; BMI: Body mass index; HDL – High-density lipoprotein; BP – Blood pressure.

and lost statistical significance after multivariate adjustment. In contrast, waist-to-height ratio remained highly significant, with an adjusted odds ratio exceeding 140, reflecting an exceptionally strong and independent association with early prediabetes/T2DM. Numerous studies support the superiority of waist-to-height ratio and waist circumference over BMI in predicting adiposity and cardiometabolic risk in youth [27–29]. Consistent with our findings, Brambilla et al. demonstrated

that waist-to-height ratio explained a greater proportion of variance in percent body fat (64%) than BMI (32%) among U.S. children and adolescents, with its predictive power increasing to 80% after adjusting for age and sex [27]. Similarly, Nambiar et al. (2010) found that waist-to-height ratio effectively identified Australian youth with elevated body fat and adverse cardiometabolic profiles, including higher triglycerides and lower HDL cholesterol [28]. These findings reinforce the growing evidence that waist-to-height ratio is a more reliable indicator of cardiometabolic risk than BMI in pediatric populations, particularly in detecting early signs of conditions such as prediabetes and T2DM.

Interestingly, lifestyle and dietary variables, including sedentary behavior, physical activity, total energy intake, and sugar consumption, did not persist as independent predictors of prediabetes/T2DM after multivariate adjustment. This finding contrasts with prevailing narratives linking the global rise in obesity to adverse lifestyle factors such as physical inactivity and the overconsumption of energy-dense, ultra-processed foods [30–32]. This likely reflects that adiposity mediates much of the effect of lifestyle on glycemic outcomes, and that single-point self-reported behaviors may not adequately capture the cumulative exposure needed to influence early glycemic status. Furthermore, race and ethnicity were not significant predictors in this dataset, contrasting with prior reports of elevated risk in minority adolescents [4]. This may indicate that when obesity and central adiposity are accounted for, the independent effect of race on prediabetes/T2DM diminishes, or that the 2021–2023 NHANES cycle sample sizes limited subgroup detection power.

## Implications

From a clinical and public health perspective, these findings suggest that screening strategies relying solely on BMI may miss high-risk adolescents. Integrating waist-to-height ratio into routine pediatric assessment could enhance early identification, particularly among males, and inform targeted interventions to reduce central adiposity before glycemic deterioration occurs.

## Strengths and limitations

This study has notable strengths, including the use of recent, nationally representative data with standardized anthropometric and biochemical assessments, and a robust multivariable regression framework to isolate independent predictors. By directly contrasting BMI and waist-to-height ratio, the analysis contributes new evidence supporting waist-to-height ratio as a superior anthropometric marker for adolescent with prediabetes/T2DM in the U.S.

However, several limitations merit consideration. First, the cross-sectional design precludes causal inference between risk factors and prediabetes or T2DM. Second, the reliance on single-point fasting glucose and HbA1c measurements may misclassify glycemic status, as adolescents often exhibit variability. Third, dietary and activity measures are self-reported and may underestimate true exposure. Finally, the very wide confidence intervals for waist-to-height ratio indicate that while the association is strong, precision is limited, likely due to sparse events in certain subgroups, and should be interpreted cautiously.

## Conclusion

Nearly 1-in-3 American adolescents has prediabetes or T2DM. Male gender and younger age showed increased risk. The findings underscore that central adiposity, specifically measured by waist-to-height ratio, is a superior and independent predictor of prediabetes/T2DM compared to general overweight/obesity (BMI). This highlights the critical need for early screening and targeted prevention strategies that incorporate waist-to-height ratio into routine pediatric assessment, focusing on central adiposity and demographic risk factors.

## Supporting information

**S1 Data. Imputed Stata dataset used for the analysis of prediabetes and diabetes among U.S. adolescents (NHANES 2021–2023).**
(DTA)

**S1 Table. Weighted prevalence of prediabetes and diabetes among US adolescents (10–19 years) in the NHANES fasting subsample (n = 571), stratified by key demographic and clinical variables.**
(DOCX)

**S2 Table. Sequential Logistic Regression Models Examining the Association Between Overweight/Obesity and Abnormal Glucose Status Among U.S. Adolescents (NHANES 2021–2023, MI = 20).**
(DOCX)

## Author contributions

**Conceptualization:** Eric Peprah Osei.

**Data curation:** Eric Peprah Osei.

**Formal analysis:** Eric Peprah Osei.

**Investigation:** Eric Peprah Osei.

**Methodology:** Eric Peprah Osei.

**Software:** Eric Peprah Osei.

**Validation:** Eric Peprah Osei.

**Visualization:** Eric Peprah Osei.

**Writing – original draft:** Eric Peprah Osei.

**Writing – review & editing:** Eric Peprah Osei.

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
