## [Decision Letter · Decision Letter 0]

30 Oct 2025

PGPH-D-25-02293

Prevalence and predictors of prediabetes/type 2 diabetes in adolescents in the United States: Data from NHANES (2021-2023)

Dear Dr. Peprah Osei,

Thank you for submitting your manuscript to PLOS Global Public Health. After careful consideration, we feel that it has merit but does not fully meet PLOS Global Public Health’s publication criteria as it currently stands. Therefore, we invite you to submit a revised version of the manuscript that addresses the points raised during the review process.

We look forward to receiving your revised manuscript.

Kind regards,

Doreen Larvie, Ph.D

Academic Editor

Journal Requirements:

1. Please ensure that your Ethics Statement is available in its entirety at the beginning of your Methods section, under a subheading 'Ethics Statement'. It must include:

1) The name(s) of the Institutional Review Board(s) or Ethics Committee(s)

2) The approval number(s), or a statement that approval was granted by the named board(s)

3) (for human participants/donors) - A statement that formal consent was obtained (must state whether verbal/written) OR the reason consent was not obtained (e.g. anonymity).

2. Please provide separate figure files in .tif or .eps format.

Additional Editor Comments (if provided):

Reviewers' comments:

Reviewer's Responses to Questions

**Comments to the Author**

1. Does this manuscript meet PLOS Global Public Health’s publication criteria?

Reviewer #1: Yes

Reviewer #2: Yes

2. Has the statistical analysis been performed appropriately and rigorously?

Reviewer #1: Yes

Reviewer #2: Yes

3. Have the authors made all data underlying the findings in their manuscript fully available (please refer to the Data Availability Statement at the start of the manuscript PDF file)?

Reviewer #1: Yes

Reviewer #2: Yes

4. Is the manuscript presented in an intelligible fashion and written in standard English?

Reviewer #1: Yes

Reviewer #2: Yes

Reviewer #1: Thank you for the opportunity to review this manuscript. The manuscript is well written and provides a timely update on risk factors for prediabetes and diabetes among a nationally representative sample of adolescents in the United States. I have the following comments for consideration prior to publication:

Methods:

1) While I agree with the author that categorizing glucose status into “normal” and “prediabetes/diabetes” status allows for a more efficient analysis, I do not agree with the author’s statement that is allows for a more thorough analysis. In fact, I think grouping them in this way could hide meaningful differences in risk factors for prediabetes versus diabetes.

2) In a similar vein, I understand the purpose of grouping prediabetes/diabetes into one outcome due to the relatively small sample size. While separating the outcomes into “prediabetes” and “diabetes” would result in reduced statistical power, it could be an interesting sensitivity analysis to see if there are any unique risk factors for prediabetes and diabetes, respectively.

3) The author categorizes energy intake into four quartiles (<1396KCAL, 1396-1837.9KCAL, 1838-2388.9KCAL, >=2389 KCAL). Was there any consideration for whether the energy intake levels were appropriate for an individual’s age/sex? Perhaps the author could consider creating a category of consuming less than recommended KCAL, consuming recommended KCAL, consuming more than recommended KCAL, based on age and sex.

4) In the clinical and biochemical factors paragraph, the author writes “elevated BP is defined as SBP of 120-129 mmHg and DBP < 80mmHg”. This seems incorrect.

5) In the final paragraph of the methods, the author reports on collinearity and variance inflation factors. This should be in the results section under a “modeling assumptions” subsection.

Results:

1) The author relies too heavily on p-values

2) The author provides p-values in Table 1 to evaluate whether there are statistically significant differences in baseline demographics. However, the use of p-values to evaluate such differences should be avoided and removed in alignment with guidelines from the American Statistical Association (https://amstat.tandfonline.com/doi/pdf/10.1080/00031305.2016.1154108 ) and the ICMJE (http://www.icmje.org/recommendations/browse/manuscript-preparation/preparing-for-submission.html).

3) Do not bold statistically significant results in a table, this can be misleading and result in readers overlooking potentially important or interesting findings.

4) The author reports that table 2 represents “weighted” prevalence but weighting is never discussed in the methods section. Please clarify how results were weighted.

Additional comment from editor: The methods section should clearly indicate whether NHANES survey weights and design variables (sampling weights, strata, PSUs) were used in the analysis. If not, please justify this approach, as unweighted analyses may not yield representative or unbiased results.

5) When reporting results and stating differences between groups please report the prevalence rates for each group rather than the p-value to indicate meaningful difference. For example, “…adolescents with overweight/obesity and those with abdominal obesity showed significantly higher prevalence rates (36.4% and 36.7%, respectively” than their healthier-weight counterparts (X%)”. The p-value does not tell us anything about how clinically meaningful the prevalence difference is.

6) Similarly, when reporting results of logistic regression analysis please report confidence intervals instead of p-values.

7) It is interesting that the effect of BMI status reverses direction in the multivariate analysis (though the confidence interval overlaps with the univariate CI). The author should consider whether there is the potential for collider bias (https://jamanetwork.com/journals/jama/fullarticle/2790247) in the multivariate regression.

Reviewer #2: This study analyzed data from 1,998 U.S. adolescents (10–19 years) from the National Health and Nutrition Examination Surveys (NHANES, 2021–2023) to determine the prevalence and predictors of prediabetes/type 2 diabetes.

Key Findings

The overall weighted prevalence of prediabetes or diabetes was a concerning 30.8%, meaning nearly 1-in-3 American adolescents has the condition.

Prevalence Disparities:

Prevalence was significantly higher in males (38.1%) compared to females (23.4%).

Mexican American (33.6%) and Other Hispanic (33.3%) adolescents had the highest rates across racial/ethnic groups.

Adolescents with overweight/obesity (36.4%) and abdominal obesity (waist-to-height ratio \ge 0.5) (36.7%) showed significantly higher prevalence.

Predictors of Prediabetes/Diabetes

Univariate Analysis found several significant associations:

Lower odds: Older age (OR=0.93, p=0.045) and female gender (OR=0.50, p=0.001).

Higher odds: Overweight/obesity (OR=1.57, p=0.012), elevated waist-to-height ratio (OR=24.04, p=0.002), total daily sugar intake (OR=1.003, p=0.042), low \text{HDL cholesterol} (\le 45 \text{ mg/dL}) (OR=1.41, p=0.032), higher systolic BP, and higher diastolic BP.

Multivariate Analysis (after adjusting for confounders) identified three independent, significant predictors:

Elevated Waist-to-Height Ratio (Central Adiposity): This emerged as the strongest independent predictor, with adolescents having over 146 times higher odds (AOR=146.19, p=0.004). BMI status, in contrast, lost significance in the multivariate model.

Female Gender: Associated with lower odds (AOR=0.52, p=0.002) compared to males.

Older Age: Associated with lower odds (AOR=0.91, p=0.025).

Conclusion and Implications

Nearly 1-in-3 American adolescents has diabetes or prediabetes. Male gender and younger age showed increased risk. The findings underscore that central adiposity, specifically measured by waist-to-height ratio, is a superior and independent predictor of prediabetes/diabetes compared to general overweight/obesity (BMI). This highlights the critical need for early screening and targeted prevention strategies that incorporate waist-to-height ratio into routine pediatric assessment, focusing on central adiposity and demographic risk factors

**Do you want your identity to be public for this peer review?** For information about this choice, including consent withdrawal, please see our Privacy Policy

Reviewer #1: No

Reviewer #2: No

---

## [Editor Report · Decision Letter 1]

19 Nov 2025

Prevalence and predictors of prediabetes/type 2 diabetes in adolescents in the United States: Data from NHANES (2021-2023)

PGPH-D-25-02293R1

Dear Mr Peprah Osei,

We are pleased to inform you that your manuscript 'Prevalence and predictors of prediabetes/type 2 diabetes in adolescents in the United States: Data from NHANES (2021-2023)' has been provisionally accepted for publication in PLOS Global Public Health.

Best regards,

Doreen Larvie, Ph.D

Academic Editor